# Biofunctionalized Nanostructured Yttria Modified Non-Invasive Impedometric Biosensor for Efficient Detection of Oral Cancer

**DOI:** 10.3390/nano9091190

**Published:** 2019-08-22

**Authors:** Suveen Kumar, Shweta Panwar, Saurabh Kumar, Shine Augustine, Bansi D. Malhotra

**Affiliations:** 1Nanobioelectronics Laboratory, Department of Biotechnology, Delhi Technological University, Delhi 110042, India; 2Department of Chemistry, University of Delhi, Delhi 110007, India

**Keywords:** non-invasive, dielectric constant, yttrium oxide, metal oxide, nanoparticles, CYFRA-21-1

## Abstract

We report results of the studies relating to the development of an efficient biosensor for non-invasive detection of CYFRA-21-1 cancer biomarker. We used a low dielectric constant material (nanostructured yttrium oxide, nY_2_O_3_) for the fabrication of the biosensing platform. The nY_2_O_3_ was synthesized via solvothermal process and functionalized using 3-aminopropyl triethoxy silane (APTES). Electrophoretic deposition (EPD) of the functionalized nanomaterial (APTES/nY_2_O_3_) onto an indium tin oxide (ITO)-coated glass electrode was conducted at a DC potential of 50 V for 60 s. The EDC-NHS chemistry was used for covalent immobilization of −COOH bearing monoclonal anti-CYFRA-21-1 onto −NH_2_ groups of APTES/nY_2_O_3_/ITO electrode. To avoid the non-specific interaction on the anti-CYFRA-21-1/APTES/nY_2_O_3_/ITO immunoelectrode, bovine serum albumin (BSA) was used. X-ray diffraction (XRD), transmission electron microscopy (TEM), and field emission scanning electron microscopy (FESEM) were utilized for structural and morphological studies, whereas Fourier-transform infrared spectroscopy (FTIR) was used for the bonding analysis. Cyclic voltammetry (CV) and electrochemical impedance spectroscopy (EIS) techniques were used for electrochemical characterization and response studies of fabricated electrodes. The fabricated immunosensor (BSA/anti-CYFRA-21-1/APTES/nY_2_O_3_/ITO) exhibited linearity in the range of 0.01–50 ng·mL^−1^, sensitivity of 226.0 Ω·mL·ng^−1^, and lower detection limit of 0.01·ng·mL^−1^. A reasonable correlation was observed between the results obtained using this biosensor and concentration of CYFRA-21-1 measured through ELISA (enzyme-linked immunosorbent assay) technique in salivary samples of oral cancer patients.

## 1. Introduction

Cancer is an abnormal and uncontrolled cell growth [1,2]. According to a WHO report, cancer is currently the secondmost death-causing disease worldwide and about 8.8 million deaths were reported in 2015 due to cancer [3]. Lung, colon, breast, prostate, oral, and ovarian cancers are the most prominent cancers in the present-day world. Oral cancer (OC) is the sixthmost death-causing cancer and it can occur in lips, cheeks, gingiva, or palate part in the mouths of human beings. The main reason behind oral cancer is the mutation, which occurs at the gene level due to which uncontrolled cell cycle and differentiation do not take place [4]. Mutations in a gene may occur due to the use of a mutagenic agent which is widely present in tobacco, cigarettes, cigars, alcohol, etc. [5]. The current detection techniques such as biopsy, cytopathology, visualization adjuncts, etc. can be used for detection of OC at advanced stages [6,7,8,9]. Moreover, these techniques require a tissue sample for analysis, and hence are invasive, time-consuming, expensive, and require trained experts for sample collection as well as data analysis. To overcome these limitations, non-invasive [10] biosensors can play an important role since they offer many advantages such as painless detection, high sensitivity, cost-effectiveness, and require low sample volume.

For a non-invasive biosensor, salivary CYFRA-21-1 biomarker can be used for OC detection [11,12]. Kumar et al. recently used a number of materials including zirconia, hafnia, and nanocomposites for development of non-invasive biosensors. Tiwari et al. used lanthanum hydroxide for the fabrication of a biosensor for CYFRA-21-1 detection via CV or DPV (differential pulse voltammetry) techniques [13,14,15,16,17,18]. In these methods, even a slight change in volume of a biological sample and analytical solution can adversely affect the peak current response. Besides this, the accurate determination of active surface area of the immunoelectrode is an important issue. These limitations can, perhaps, be overcome by using an EIS technique.

In the fabrication of a biosensor, nanostructured metal oxides (NMOs), nanostructured metal sulfides (NMSs), reduced graphene oxide, carbon nanotubes, etc. have been extensively used as immobilization matrices [19,20,21,22,23,24,25,26]. The nanostructured oxides of metals (e.g., magnesium oxide, zinc oxide, molybdenum trioxide, hafnia, titania, zirconia, etc.) have been found to have fascinating nano-morphological, functional, biocompatible, and electrochemical properties, and enhanced electron-transfer kinetics [19,27,28,29,30,31,32,33]. Among these NMOs, the nano-sized yttrium oxide (Yattria, nY_2_O_3_) exhibits high surface-to-volume ratio, fast oxygen ion mobility, efficient charge transfer ability, chemical inertness, sharp line emission bands, and biocompatibility [34,35,36]. In addition, it is known to have high quantum yield, excellent photo stability, and low dielectric constant (k), making it a potential candidate for application towards the development of biosensors [37,38,39,40,41]. The low dielectric constant (13) of yttrium oxide makes the thin film highly conductive [42]. Besides this, the oxygen moieties in Y_2_O_3_ can facilitate functionalization and covalent immobilization of antibodies. Rasheed et al. developed yttria-reduced graphene oxide nanocomposite based genosensor for detection of BRCA 1 (breast cancer) gene [40]. Efforts have also been made to use yttria–zirconia nanocomposite for the fabrication of sensors that can be used for detection of hydrogen, oxygen, and nitric oxide [41,43].

In this paper, we report results of the systematic studies relating to the development of a low dielectric material (nY_2_O_3_)-based electrochemical impedometric biosensor for detection of the salivary CYFRA-21-1 biomarker for OC detection. This immunosensor (BSA/anti-CYFRA-21-1/APTES/nY_2_O_3_/ITO) exhibits higher linear range (0.01–50 ng·mL^−1^) with remarkable sensitivity (226.0 Ω·mL·ng^−1^).

## 2. Experimental Section

### 2.1. Preparation and Functionalization of Nanostructured Yttrium Oxide (nY_2_O_3_)

Solvothermal synthesis process of nanoparticles was used for the synthesis of nY_2_O_3_. Initially, we separately prepared 4 mM of yttrium (III) nitrate hexahydrate, 2 mM of CTAB, and 80 mM of urea in Milli-Q water. The CTAB solution was added dropwise into yttrium (III) nitrate hexahydrate solution at 40 °C with constant stirring of 300 rpm for 2 h. Subsequently, urea was added dropwise into this solution at the same temperature and stirring conditions. Further, the whole solution was transferred to a hot (80 °C) ultrasonication bath for 2 h. The obtained precipitate was washed three times with Milli-Q water, followed by ethanol with the help of a centrifuge at 4500 rpm for 30 min, after which the precipitate was kept in a hot air oven at 80 °C to make it moisture-free. The dried product was transferred into the muffle furnace at 800 °C for 2 h. The obtained white product was crushed usinga pestle and mortar and was stored in a desiccator at room temperature.

For functionalization of nY_2_O_3_ with APTES molecules, 100 mg of nY_2_O_3_ was dissolved in a minimal amount of isopropanol and then sonicated for 30 min. The solution was kept at constant stirring (300 rpm) at 50 °C until the nanoparticles were completely dispersed, after which we added 1 mL of APTES dropwise and 5 mL of Milli-Q water. This solution was kept in a magnetic stirrer at 200 rpm at 40 °C for 48 h. Subsequently, the solution mixture was washed with Milli-Q water (4500 rpm for 15 min) to remove any unbound APTES molecules. After centrifugation, the obtained powder was kept in a hot air oven at 50 °C until a moisture-free product was obtained.

### 2.2. Fabrication of Impedometric Biosensing Platform

A colloidal suspension of functionalized nanomaterial (APTES/nY_2_O_3_; concentration of 0.5 mg·mL^−1^) in acetonitrile was prepared via ultrasonication. Thereafter, 10 mL of colloidal solution was poured into a two-electrodes system (platinum as reference and ITO as working electrode), kept 1 cm apart to each other. Further, the optimized DC potential of 50 V was applied in the two-electrodes system using an electrophoretic deposition (EPD) unit for 60 s. The uniform thin layer of APTES/nY_2_O_3_ was obtained on ITO electrode.

Immobilization of the monoclonal antibodies (anti-CYFRA-21-1) was carried onto APTES/nY_2_O_3_/ITO electrode via EDC-NHS coupling chemistry. For this purpose, we took 15 µL of anti-CYFRA-21-1 and mixed it with 7.5 µL of 0.4 M of EDC and 7.5 µL of 0.1 M NHS, and uniformly drop cast onto APTES/nY_2_O_3_/ITO electrode. This process helped in covalent bond formation between −COOH and −NH_2_ group present on anti-CYFRA-21-1 and APTES/nY_2_O_3_/ITO electrode, respectively. Subsequently, the fabricated platform (anti-CYFRA-21-1/APTES/nY_2_O_3_/ITO) was washed with PBS to remove any unbound antibodies. For blocking of non-specific binding sites of the prepared electrode, we spread 30 mL of BSA (1 mg·dL^−1^) molecule. Finally, the fabricated immunoelectrodes (BSA/anti-CYFRA-21-1/APTES/nY_2_O_3_/ITO) were stored at 4 °C. Scheme 1 shows the stepwise fabrication process of the BSA/anti-CYFRA-21-1/APTES/nY2O3/ITO biosensor.

### 2.3. Biocompatibility Studies of nY_2_O_3_

The biocompatibility study of nY_2_O_3_ was carried out on HEK 293 cell line by using MTT colorimetric assay technique. In this technique, the yellow color of MTT dye got converted into purple color of the formazan product by mitochondrial succinate dehydrogenase enzyme present in the live cells. The colored formazan product was solubilized in a buffer, and further, intensity of color was measured at 540 nm by using an ELISA plate reader.

For biocompatibility assay, 96-well plates were used. First of all, we seeded the HEK 293 cells (cell density as 105 cells per well) in a 96-well plate and incubated it at 37 °C in a humidified 5% CO_2_ incubator. After the cells attained 70% confluence, we added 10 mg·mL^−1^ to 250 mg·mL^−1^ of nY_2_O_3_ and again incubated it for 48 h in a CO_2_ incubator. For the MTT assay, 200 mL of MTT solution (0.5 mg mL^−1^ in DMEM) was added to each well and incubated for 2h. Further supernatant aspirated and the buffer (100 mL isopropanol containing 0.06 M HCl and 0.5% SDS) were added to each well to solubilize the formazan crystals. With the help of an ELISA plate reader, we measured the absorbance of colored product obtained at 540 nm. The untreated cells were taken as control with 100% viability and wells with MTT reagent without cells were used as blank to calibrate the spectrophotometer to zero absorbance. The relative cell viability (%) compared to control cells was calculated using [abs]_sample_/[abs]_control_ × 100. All the experiments were carried out in triplicate.

## 3. Results and Discussion

### 3.1. Structural and Morphological Studies

To investigate the structural and morphological characteristics of synthesized nY_2_O_3_ and the modified ITO electrodes, we conducted XRD, FESEM, and TEM studies. The obtained XRD pattern of the material is shown in Figure 1a. The diffraction peaks found at 2θ angle of 20.5, 29.2, 33.9, 40.0, 43.6, 48.5, 57.7, and 78.7 corresponded to the planes of (211), (222), (400), (332), (134), (440), (622), and (662), respectively. These peaks are in good correlation with the JCPDS NO. 65-3178, indicating successful synthesis of Y_2_O_3_ with a crystalline size (D) ~80 nm, calculated using the Debye Scherer method [14]. To confirm the size of the synthesized nanostructured material, we carried out TEM analysis at different magnifications (Figure 1b–d) and found that the synthesized nanomaterial was oval-shaped and had an average size of ~80 nm. The FESEM technique (Figure 2) was used at different magnifications to investigate the morphology at the surface of (i) nY_2_O_3_, (ii) electrophoretically deposited APTES/nY_2_O_3_, and (iii) anti-CYFRA-21-1/APTES/nY_2_O_3_ electrodes. The images (Figure 2a–c) observed at different magnifications confirm the presence of nanosized yttria, and these results are in agreement with the results of XRD and TEM studies. Further, in Figure 2d–f, the uniform deposition of APTES/nY_2_O_3_ onto the ITO electrode can be seen. Interestingly, the globular morphology with uniform pattern (dendritic-like) was observed (Figure 2e–g) after immobilization of the anti-CYFRA-21-1.

### 3.2. Fourier Transform-Infrared (FTIR) Spectroscopy Study

To investigate the presence of functional groups and bonds at the surfaces of APTES/nY_2_O_3_/ITO and anti-CYFRA-21-1/APTES/nY_2_O_3_/ITO electrodes, FTIR studies were conducted and the results are shown in Figure 3a,b, respectively. In Figure 3a, the band present at 1340 cm^−1^ is due to the presenceof C–N, and 1530 cm^−1^ is assigned to the presence of N–H (primary and secondary) bond present in amine molecules of the APTES-functionalized nY_2_O_3_ [44]. The bands present between 3000 to 4000 cm^−1^ are attributed to the N–H stretching of amine, C–H molecules present in APTES, and O–H molecules present in PBS [44]. These results confirm the presence of amine (–NH_2_) group onto the APTES/nY_2_O_3_/ITO electrode. Further, we investigated the formation of amide bond after the immobilization of anti-CYFRA-21-1 onto APTES/nY_2_O_3_/ITO electrode (Figure 3b). In Figure 3b, the presence of band at 1260 cm^−1^ confirms the formation of C–N (amide) bond between −COOH group present at the F_c_ region of the anti-CYFRA-21-1 and −NH_2_ groups of APTES/nY_2_O_3_/ITO electrode [44]. The bands seen at 1560 cm^−1^ and 1632 cm^−1^ indicate the presence of N–H (primary and secondary) bending and C=O groups of amides. Bands found at 1055 cm^−1^ and 1095 cm^−1^ are attributed to the C–O bands of carboxylic acid group, and bands present at 3350 cm^−1^ are assigned to the presence of N–H band of amines and O–H groups of PBS [44]. The FTIR results confirm the covalent immobilization (amide bond formation) of anti-CYFRA-21-1 onto the APTES/nY_2_O_3_/ITO electrode.

### 3.3. Biocompatibility Studies

The biocompatibility studies of nanomaterials are known to provide an insight towards the method of immobilization and facilitate the selection of nanomaterial for development of implantable chip [15]. Besides this, biocompatible nanomaterials can be helpful in providing a favorable environment for immobilized biomolecules. We performed the biocompatibility study through MTT assay and the results are shown in Figure 4 and Appendix A. The cell viability of HEK 293 cells was more than 95% upto 50 µg mL^−1^ concentration of nY_2_O_3_, after which the cell viability was found to be decreased. These results indicated that the synthesized nanomaterial (nY_2_O_3_) was favorable for the immobilization of biomolecules to be utilized for the fabrication of biosensors.

### 3.4. Electrochemical Characterization

Electrochemical investigations were carried out via EIS and CV studies using a redox mediator [Fe(CN)_6_]^3−/4−^(5 mM) in PBS. The effect of pH on the BSA/anti-CYFRA-21-1/APTES/nY_2_O_3_/ITO immunoelectrode was conducted via CV. The maximum activity of the immunoelectrode was observed at pH 7.0 [{Fe(CN)_6_}^3−/4−^(5 mM) in PBS buffer]. In both acidic and basic pH, the peak current decreased (Figure 5i). This result was ascribed to the maximum activity of anti-CYFRA-21-1 at pH 7.0 of PBS. In both acidic and basic pH, the antibodies became denatured, resulting in a decrease of the peak current [14,45].

The electrochemical studies of the ITO, APTES/nY_2_O_3_/ITO, anti-CYFRA-21-1/APTES/nY_2_O_3_/ITO, and BSA/anti-CYFRA-21-1/APTES/nY_2_O_3_/ITO immunoelectrodes were investigated via CV. The peak current of the hydrolyzed ITO and APTES/nY_2_O_3_ electrodes was found to be ~0.9 mA and ~0.7 mA, respectively (Figure 5ii). After immobilization of the antibodies and BSA, the peak current decreased to ~0.68 mA and ~0.66 mA, respectively. This decrease in peak current was attributed to the insulating properties of the biomolecules (anti-CYFRA-21-1 and BSA) [46]. The results of scan rate studies of the APTES/nY_2_O_3_/ITO and BSA/anti-CYFRA-21-1/APTES/nY_2_O_3_/ITO electrodes carried out from 50 to 150 mV/s revealed that the fabricated immunoelectrode showed uniform electrochemical behavior (Figure 5iii,iv).

The cathodic and anodic peak current increased linearly towards the higher positive and lower negative potential, respectively (Inset i, Figure 5iii,iv and Appendix A). The observed linearity with respect to the scan rate revealed that the fabricated electrodes exhibited diffusion control process, indicating that we can use scan rate in the range, 50 to 150 mV/s for the desired electrochemical studies. Equations (1)–(4) well describe the linearity obtained between cathodic and anodic peak current response with respect to scan rate [13,30].
I_1_ = [3.53 μA(s mV^−1^) × (scan rate[Mv s^−1^])] + 562.91 μA, R^2^ = 0.987(1)
I_2_ = −[2.62 μA(s mV^−1^) × (scan rate[mV s^−1^])] − 520.36 μA, R^2^ = 0.985(2)
I_3_ = [3.47 μA(s mV^−1^) × (scan rate[mV s^−1^])] + 542.9 μA, R^2^ = 0.989(3)
I_4_ = −[2.52 μA(s mV^−1^) × (scan rate[mV s^−1^])] − 465.64 μA, R^2^ = 0.987(4)
where I_1_ and I_3_ are the cathodic peak current of APTES/nY_2_O_3_/ITO and BSA/anti-CYFRA-21-1/APTES/nY_2_O_3_/ITO; I_2_ and I_4_ are the anodic peak current of APTES/nY_2_O_3_/ITO and BSA/anti-CYFRA-21-1/APTES/nY_2_O_3_/ITO electrode, respectively. Furthermore, it was observed that the difference between the peak potential of the cathode and anode of the APTES/nY_2_O_3_/ITO and BSA/anti-CYFRA-21-1/APTES/nY_2_O_3_/ITO electrodes exhibited a linear relationship with respect to the scan rate (Inset ii, Figure 5 iii,iv) and followed Equations (5) and (6).
∆E_1_ = [0.006 V(s/mV) × (scan rate[mV/s])] + 1.08 V, R^2^= 0.987(5)
∆E_2_ = [0.006 V(s/mV) × (scan rate[mV/s])] + 1.01 V, R^2^= 0.988(6)
where ∆E_1_ and ∆E_2_ reveal differences in peak potential of APTES/nY_2_O_3_/ITO and BSA/anti-CYFRA-21-1/APTES/nY_2_O_3_/ITO, respectively.

The diffusion coefficient “D” of the electrode was calculated via RandlesSevick equation [17,47]:I_p_ = (2.69 × 10^5^) n^3/2^AD^1/2^Cυ^1/2^(7)
where I_p_ is the peak current of the electrode, n is the number of electrons transferred in the redox reaction (n = 1), A is the surface area of the electrode (0.25 cm^2^), C is the concentration of the redox species (5 × 10^−3^ mol·cm^−2^), and υ is the scanning rate (50 mV·s^−1^). The “D” value at each step of electrode fabrication was determined to be 9.68 × 10^−11^ cm^2^·s^−1^(APTES/nY_2_O_3_/ITO), 9.18 × 10^−11^ cm^2^·s^−1^ (anti-CYFRA-21-1/APTES/nY_2_O_3_/ITO), and 8.55 × 10^−11^ cm^2^·s^−1^(BSA/anti-CYFRA-21-1/APTES/nY_2_O_3_/ITO), respectively. The diffusion coefficient of the fabricated immunoelectrode was found to be lower as compared to the value of diffusion coefficient as 1.12 × 10^−^^3^ cm^2^/s [13], 2.12 × 10^−^^3^ cm^2^/s [15], and 0.62 × 10^−^^3^ cm^2^/s [16]) for reported immunoelectrodes, indicating that the nY_2_O_3_ was a more efficient nanomaterial for the fabrication of a cancer biosensor.

Quantification of antibodies onto the fabricated biosensing electrode surface are known to provide a better insight into the functioning of a biosensor, and it can be calculated by using Brown-Anson equation (Equation (8)) [16].
I_p_ = n^2^F^2^γAυ(4RT)^−1^(8)
where I_p_ represents the peak current; A is the surface area of the electrode (we have taken 0.25 cm^2^ active surface area for performing the electrochemical characterization and response studies); ν is the scan rate (V/s); γ is the surface concentration of the absorbed electro-active species; F is the Faraday constant; R is the gas constant; and T is room temperature.The number of antibodies immobilized onto the fabricated immunoelectrode was found to be 5.91 × 10^−8^ mol/cm^2^, which was higher as compared to the reported biosensing platforms for CYFRA-21-1 detection [16]. Kumar et al. [13] indicating that nY_2_O_3_ provided better conjugation of biomolecules, leading to enhanced biosensing characteristics.

### 3.5. Electrochemical Impedometric Response, Control, Interferent, Reproducibility,and Shelf Life Studies

EIS is an important electrochemical technique that can be used to investigate the dynamics of antigen–antibodies interactions. We conducted EIS measurements using a three-electrode system in the frequency range from 0.01 to 10^5^ Hz at a potential of 0.10V. The impedometric response obtained between fabricated immunoelectrode and CYFRA-21-1 can be understood by using Nyquist plot. In this plot, impedance can be expressed by the real part (Z’) or the modules |Z| and the phase shift φ with respect to the imaginary part (Z”) determined by the Randles circuit [48]. In the Randles circuit, R_ct_ is the charge transfer resistance, C_dl_ is capacitance of the dielectric layer, and Z_w_ is Warburg impedance [48]. As shown in Figure 6a, we plotted the Nyquist as a function of CYFRA-21-1 concentration. It was observed that the diameter of the semicircle increased with increasing concentration of CYFRA-21-1 (Figure 6a). We also drew the linearity graph between the R_ct_ value obtained from Randles circuit and the concentration of CYFRA-21-1 (Figure 6b). With increasing concentration, the R_ct_ value increased linearly from the log of 0.01 ng·mL^−1^ to 50 ng·mL^−1^. This increase in R_ct_ was attributed to the insulating characteristics of theCYFRA-21-1 layer formed onto the BSA/anti-CYFRA-21-1/APTES/nY_2_O_3_/ITO immunoelectrode [49,50]. As the number of CYFRA-21-1 molecules increased, the F_ab_ region of anti-CYFRA-21-1 present on the immunoelectrodes became saturated due to binding of the paratope of CYFRA-21-1 on the antibodies. When the F_ab_ region of the immunoelectrode was saturated, CYFRA-21-1 did not bind with the immunoelectrode, hence we did not find any change in the R_ct_ value [49,50]. The linearity graph (Equation (9)) indicated that the fabricated immunoelectrode could be utilized to measure the concentration of CYFRA-21-1 from 0.01 to 50 ng·mL^−1^.
I_p_ = [(0.23 KΩ·mL·ng^−1^) × log {conc. of CYFRA-21-1 (ng mL^−1^)}] + 1.59, R^2^ = 0.94(9)

A control study was conducted via EIS to investigate the interaction of theAPTES/nY_2_O_3_/ITO electrode with increasing CYFRA-21-1 concentration from 0.01 ng·mL^−1^ to 50 ng·mL^−1^. The obtained R_ct_ values through Randles circuit were plotted as a function of CYFRA-21-1 concentration (Figure 6c). We did not observe any significant change in the R_ct_ value due to the APTES/nY_2_O_3_/ITO electrode as a function of different concentrations of CYFRA-21-1 biomolecules. The observed electrochemical response revealed that the CYFRA-21-1 biomolecules did not interact with the APTES/nY_2_O_3_/ITO electrode. The observed sensing characteristics were due to the interaction between antibodies immobilized onto the biosensing platform (BSA/anti-CYFRA-21-1/APTES/nY_2_O_3_/ITO) and CYFRA-21-1 biomolecules.

We investigated the effect of various interferents present in real saliva samples of OC patients onto the EIS response of the fabricated biosensing electrode. In saliva samples, many analytes such as ET-1, CEA, glucose, NaCM cellulose, NaCl, Kcl, CaCl_2_, cTn-I, etc. are known to be present [14]. We added these analytes into the PBS solution one-by-one, after which the EIS studies were conducted (Figure 6d). We did not find any significant changes on addition of analytes. The obtained results indicate that the fabricated immunoelectrode exhibited the EIS response towards CYFRA-21-1 biomolecules only. We checked the reproducibility of the BSA/anti-CYFRA-21-1/APTES/nY_2_O_3_/ITO biosensing electrode. In this study, we took eight different fabricated biosensing electrodes and measured both CV and EIS response. We found that deviation in the electrochemical response. The average percentage relative standard deviation (%RSD) was 5.4% and 2.1% in the CV and EIS response, respectively. The obtained results are shown in Appendix A. The shelf life of the developed biosensing electrode was measured by EIS in ferro-ferricynide containing PBS buffer at regular intervals of one week to five weeks (Appendix A). We did not find any significant changes in the Nyquist plot, Randles circuit,orin R_ct_ value, indicating that the immunoelectrode can be safely used for up to five weeks.

### 3.6. Oral Cancer Patient Samples Analysis

Concentration of CYFRA-21-1 biomarker present in salivary samples of seven different OC patients was estimated using a double sandwich ELISA technique. Thirty microliters of these biological saliva samples were used to estimate the CYFRA-21-1 concentration using the biosensing electrode (BSA/anti-CYFRA-21-1/APTES/nY_2_O_3_/ITO), and results were correlated with ELISA (Appendix A). The good correlation (average %RSD = 3.92%) between CYFRA-21-1 concentration obtained via the electrochemical impedance spectroscopy and ELISA techniques revealed that the developed biosensing platform could be efficiently used for the detection of salivary CYFRA-21-1 biomarker in OC patients. We also compared the biosensing characteristics of this biosensor with those reported in literature to date (Table 1). It can be seen that the fabricated biosensor is more efficient as compared to the reported biosensors for CYFRA-21-1 detection.

## 4. Conclusions

The biocompatible and low dielectric constant nY_2_O_3_ has been synthesized via solvothermal process and has been used for the fabrication of a label-free impedometric biosensor for saliva-based cancer detection. The fabricated BSA/anti-CYFRA-21-1/APTES/Y_2_O_3_/ITO immunoelectrode can be used to detect salivary CYFRA-21-1 in the linear range from 0.01 to 50 ng·mL^−1^, sensitivity of 226.0 Ω·mL·ng^−1^, lower detection limit of 0.33 ng·mL^−1^, and shelf life of five weeks. It should be interesting to utilize the biocompatible nY_2_O_3_ for the development of point-of-care devices, both for detection of communicable and non-communicable diseases.

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
