# Peer review of "Biofunctionalized Nanostructured Yttria Modified Non-Invasive Impedometric Biosensor for Efficient Detection of Oral Cancer"

_nanomaterials, 2019, doi:10.3390/nano9091190_

Round 1
Reviewer 1 Report
This article is really good.
However, here are some comments to help the authors:
*P3 line 94: why did you add the 5mL of water; to obtain a densely packed and well defined monolayer the residual water is sufficient?
*P4 figure 1: in the legend there is confusion between magnification and the scale (nearly same problem with fig2's legend), please correct
*P4 line 163: OH are also coming from the crosslinking of APTES (see my remark about P3)
*P6 figure 3: the photographies are too small and I cannot see the cells. The magnification is not suitable for the reader
*general remarks:
figure5b: well it is log scale but you should make more comment about your linearity (R2 and so on)
Regarding the analytical performance comparison with other biosensors, you did well. Regarding to the cost of fabrication can you add some comments?
Cancer biomarkers evolve with mutation and sometime with the stage of the disease. Your specificity assay is only including molecules that are very different from your target CYFRA-21-1. The human samples that you characterized in your study are probably tested with ELISA technique to check the success of the diagnosis with your device?
Author Response
Point 1:
P3 line 94: why did you add the 5mL of water; to obtain a densely packed and well defined monolayer the residual water is sufficient?
Reply: In this work, 5 mL water was used during the functionalization of the nY2O3 (dispersed in isopropyl alcohol) through APTES linker. The addition of water helped in the hydrolysis of APTES molecules. As reported in the literature, during the silanization process, APTES molecules undergo chemical modification wherein the triethoxyl groups ((OC2H5)3-Si-(CH2)3NH2) get converted into trihydroxyl groups ((OH)3-Si-(CH2)3NH2). Further these trihydroxyl group bonds with the oxygen moieties of the metal oxide resulting in its functionalization. In the present work, 5 mL of water was found to be sufficient for binding of APTES with nY2O3.
Point 2:
P4 figure 1: in the legend there is confusion between magnification and the scale (nearly same problem with fig2's legend), please correct
Reply:
We are thankful to reviewer for the critical evaluation. We have replaced the term “magnification” with “scale” in the revised manuscript.
Point 3:
P4 line 163: OH are also coming from the crosslinking of APTES (see my remark about P3)
Reply:
The APTES molecules (Scheme R1) have the functional groups such as -OC2H5, -NH2 and –H groups and don’t possess any –OH group . Thus, –OH groups did not originate due to cross linking of APTES molecules.
Scheme R1: Schematic diagram of APTES molecules
Point 4:
P6 figure 3: the photographies are too small and I cannot see the cells. The magnification is not suitable for the reader
Reply:
As per reviewer’s suggestion, we have included modified magnified Figure 4 in the revised manuscript.
Figure (R1): (i) % Cell viability and (ii to ix) morphological view of HEK 293 cell line with various concentration of nY2O3.
Point 5:
*general remarks:
figure5b: well it is log scale but you should make more comment about your linearity (R2 and so on)
Reply:
The necessary changes have been made in the revised manuscript.
Point 6:
Regarding the analytical performance comparison with other biosensors, you did well. Regarding to the cost of fabrication can you add some comments?
Reply:
We thank the reviewer for the encouragement. Till date the available biosensing device is glucometer and the cost of the glucose sensor strip is <1$. We expect that the proposed biosensing platform is likely to cost ~1$.
Point 7:
Cancer biomarkers evolve with mutation and sometime with the stage of the disease. Your specificity assay is only including molecules that are very different from your target CYFRA-21-1.
Reply:
In the interferent studies (Figure 5d), we used other proteinaceous biomarkers that are ET-1 and CEA that are released in the saliva of the oral cancer patient. However, we thank the reviewer for the suggestion; in the next work we would explore the effect of other closely related biomarkers of CYFRA-21-1.
Point 8:
The human samples that you characterized in your study are probably tested with ELISA technique to check the success of the diagnosis with your device?
Reply:
Yes, the success of the diagnosis of oral cancer through the fabricated biosensor was validated using the enzyme linked immunosorbent assay (ELISA) technique.

Reviewer 2 Report
This paper reports on an immunosensor for detection of oral cancel biomarker molecules via simple electrochemical impedance measurements. It consists of ITO electrodes whose surface decorated with functionalized yttrium oxide nanoparticles. The functional groups on the yttrium oxide enabled to immobilize antibodies through the EDC-NHS chemistry. The authors demonstrated specific detections of cancer marker CYFRA-21-1 in clinical sample. They also evaluated the sensitivity and detection limit of the novel sensor platform.
The significance of the work lies in the evaluation of the sensor capability for human saliva samples. The experiments were also carefully designed to include many control measurements that are of crucial importance to validate the practical viability and reliability of the biosensor. The paper can be published in Nanomaterials after the authors addressed the following points in full:
The authors should add photo picture and/or illustration describing their experimental set up at the beginning of the paper to let readers see what kind of sensor it is.
In figure 2, the SEM images clearly show a drastic change in the morphology of the yttria nanostructure: the particle size seems to become smaller after being deposited on ITO; the morphology also turned out to become dendrite-like structure when adding antibody molecules. The authors should better explain what actually caused the difference here.
On Fig. 3 (a), it is explained as “The cell viability of HEK 293 cells was more than 95 % in upto 50 ugmL^-1, after which the cell viability was found to be decreased.” Meanwhile, the bar graph can be seen as a monotonic decrease in the viability with the concentration. Also, definition for the error bars need to be stated in the figure caption. Moreover, Fig. 3(b) is not discussed in the main text.
There are no Figs. 4 (c) and (d). The insets in the figure are almost impossible to read due to the too-small font size.
On page 8, the units in equations (v) and (vi) seem to be unphysical. They expressed as (s/mV) x (scan rate [mV/s])^1/2) that will result as [s/mV])^1/2. Please fix it.
On the same page, diffusion constant was shown to be 8 orders of magnitude lower than the previous literature reporting the values for different oxide materials. The authors should explain what physics underlies the huge difference in the property here.
The density of antibody molecules on the electrode surface was also found to be slightly larger compared to the previous works. Does it come from the morphologies of the nanostructure as shown in Fig. 2? It is also better described how the surface area was defined to calculate the surface density of the antibodies.
The lower detection limit is somehow not as good as the other systems. Although it may not be simply compared to the other detection techniques, it will be helpful if the authors add explanation on what possibly affect the detection limit here including the influence of electrical noise.
There are many typos throughout the manuscript. Page 3 line 8: “worked as working electrode and.”; Page 6 line 2: “The obtain results”; Page 6 line 8: “PBS buffer conducted via”; Page 8 line 36: “can be expressed e by”; Figure 5: “ct” of “Rct” needs to be lower case (the same for that in page 10) and “–Z’’ in (d) should be “Z’’.
Author Response
Point 1:
The authors should add photo picture and/or illustration describing their experimental set up at the beginning of the paper to let readers see what kind of sensor it is.
Reply:
We have incorporated Scheme R2 that shows the various fabrication steps of the biosensor in the revised manuscript.
Scheme R2: Stepwise fabrication process of fabricated BSA/anti-CYFRA-21-1/APTES/nY2O3/ITO biosensor
Point 2:
In figure 2, the SEM images clearly show a drastic change in the morphology of the yttria nanostructure: the particle size seems to become smaller after being deposited on ITO; the morphology also turned out to become dendrite-like structure when adding antibody molecules. The authors should better explain what actually caused the difference here.
Reply:
Figure 2 (a-c) shows the FESEM images of nano-powder of the synthesized nY2O3. However after functionalization with APTES and later electrophoretic deposition onto ITO electrode Figure 2 (d-e) was obtained. Change in morphology is attributed to the functionalization of material with silane and its uniform deposition of functionalized nanomaterials onto the ITO surfaces. However, after immobilization of the antibodies onto the APTES/nY2O3/ITO electrode, a globular morphology along with a uniform pattern (dendrite-like structure) was observed (Figure 2e-g) indicating successful immobilization of antibodies onto APTES/nY2O3/ITO electrode.
Point 3:
On Fig. 3 (a), it is explained as “The cell viability of HEK 293 cells was more than 95 % in upto 50 ugmL^-1, after which the cell viability was found to be decreased.” Meanwhile, the bar graph can be seen as a monotonic decrease in the viability with the concentration. Also, definition for the error bars need to be stated in the figure caption. Moreover, Fig. 3(b) is not discussed in the main text.
Reply:
For biocompatibility studies (Now Figure 4), we used MTT assay of the synthesized nanomaterials (nY2O3) onto the HEK 293 cells in triplicate. We observed (Table R1) that 95% of the HEK293 cells were viable upto 50 µg mL-1, which was adequate for development of the biosensing platform. The error bars have been calculated by repeating the same experiments in triplicates (n=3). This has been incorporated in the main text of the revised manuscript.
Table R1: % Cell viability of HEK 295 Cells with respect to various concentration of nY2O3 along with % cell viability error bar (n=3).
S. No. |
Concentration of nY2O3 (µg mL-1) |
% Cell viability (Mean value) |
% Cell viability Error bar (±) |
1. |
0 |
100.00 |
0.91 |
2. |
2 |
99.05 |
0.95 |
3. |
5 |
98.80 |
1.94 |
4. |
10 |
97.20 |
1.16 |
5. |
20 |
96.89 |
1.45 |
6. |
30 |
96.51 |
2.83 |
7. |
50 |
95.32 |
2.28 |
8. |
100 |
94.86 |
1.74 |
Point 4:
There are no Figs. 4 (c) and (d). The insets in the figure are almost impossible to read due to the too-small font size.
Reply:
We are thankful to the reviewer for pointing this typographical error. The figures have been re-labelled as Figure 5 (i), (ii), (iii) and (iv). The magnified insets of the figures have also been incorporated in the revised supplementary information [as Figure S1 (a-e)].
Point 5:
On page 8, the units in equations (v) and (vi) seem to be unphysical. They expressed as (s/mV) x (scan rate [mV/s]) ^1/2) that will result as [s/mV]) ^1/2. Please fix it.
Reply:
The desired changes in equations (v) and (vi) have been incorporated in the revised manuscript.
Point 6:
On the same page, diffusion constant was shown to be 8 orders of magnitude lower than the previous literature reporting the values for different oxide materials. The authors should explain what physics underlies the huge difference in the property here.
Reply:
Diffusion coefficient (D) can be calculated by using Randle Sevick equation i.e.
Ip= (2.69×105) n3/2AD1/2Cʋ1/2 ...............eq. R1
where Ip is the peak current of the electrode, n is the number of electrons transferred in the redox reaction (n=1), A is surface area of the electrode (0.25 cm2), C is the concentration of redox species(5×10−3 mol cm−2) and υ is the scanning rate (50 mV s−1). In all the experiments conducted for the calculation of diffusion coefficient of BSA/anti-CYFRA-21-1/nZrO2-RGO/ITO, BSA/anti-CYFRA-21-1/nHfO2/ITO, BSA/anti-CYFRA-21-1/nHfO2-RGO/ITO and present biosensing electrode i.e. BSA/anti-CYFRA-21-1/APTES/nY2O3/ITO, the value of n, A, C and ʋ are constant so diffusion coefficient ‘D’ is directly proportional to the square of the peak current of fabricated immunoelectrode i.e. Ip. As we can see in the Figure 4(ii), the peak current of BSA/anti-CYFRA-21-1/APTES/nY2O3/ITO is higher with respect to the previous work. That is perhaps the reason that it shows lower diffusion coefficient with respect to previously fabricated biosensing electrodes.
Point 7:
The density of antibody molecules on the electrode surface was also found to be slightly larger compared to the previous works. Does it come from the morphologies of the nanostructure as shown in Fig. 2? It is also better described how the surface area was defined to calculate the surface density of the antibodies.
Reply:
Morphologies of the nanostructured materials as well as number of –NH2 groups present onto the immunosensing electrode can influence the immobilization of antibodies onto the electrode surface. As discussed in the manuscript, the surface density of the antibodies has been calculated by using the Brown Anson equation (eq. i) :
Ip = n2F2γAʋ(4RT)-1...................... eq.1.
Where Ip represents the peak current, A is the surface area of the electrode (We have taken 0.25 cm2 active surface area for performing all electrochemical characterization and response studies), ν is the scan rate (V/s), γ is the surface concentration of the absorbed electro-active species, F is the Faraday constant, R is the gas constant and T is room temperature.
Point 8:
The lower detection limit is somehow not as good as the other systems. Although it may not be simply compared to the other detection techniques, it will be helpful if the authors add explanation on what possibly affect the detection limit here including the influence of electrical noise.
Reply:
Generally, the lower detection limit can be determined using two different methods
Lower detection Limit = (3 x Standard deviation)/Sensitivity Lowest value of the linear detection range
In case (i) electrical noise may influence the lower detection limit. However, in case (ii), when the experiments are conducted in triplicate then it is considered that the signal might be distributed 99% normally with 1% possibility of noise (when S/N ratio = 3). In the present work, case (ii) has been considered for determining the lower detection limit assuming that 99% of the signal (Rct values) is normalised with minimal effect of noise.
Point 9:
There are many typos throughout the manuscript. Page 3 line 8: “worked as working electrode and.”; Page 6 line 2: “The obtain results”; Page 6 line 8: “PBS buffer conducted via”; Page 8 line 36: “can be expressed e by”; Figure 5: “ct” of “Rct” needs to be lower case (the same for that in page 10) and “–Z’’ in (d) should be “Z’’.
Reply:
We are thankful to the reviews for the comments. Necessary changes have been made in the respected.

Round 2
Reviewer 2 Report
The authors have answered to all the comments raised and revised the manuscript accordingly. I now recommend publication of this paper.
Additional comment:
I still found some typos in the abstract that the authors may want to fix, such as "... an efficient biosensor for that can be ..." and "... platform comprising of ... ".
Author Response
The required corrections have been made in the revised manuscript.